# Memory Enhancement by Oral Administration of Extract of *Eleutherococcus senticosus* Leaves and Active Compounds Transferred in the Brain

**DOI:** 10.3390/nu11051142

**Published:** 2019-05-22

**Authors:** Yui Yamauchi, Yue-Wei Ge, Kayo Yoshimatsu, Katsuko Komatsu, Tomoharu Kuboyama, Ximeng Yang, Chihiro Tohda

**Affiliations:** 1Division of Neuromedical Science, Institute of Natural Medicine, University of Toyama, Toyama 930-0194, Japan; momoktt226@gmail.com (Y.Y.); kuboyama@inm.u-toyama.ac.jp (T.K.); ximeng@inm.u-toyama.ac.jp (X.Y.); 2Division of Pharmacognosy, Institute of Natural Medicine, University of Toyama, Toyama 930-0194, Japan; yueweige@aliyun.com (Y.-W.G.); katsukok@inm.u-toyama.ac.jp (K.K.); 3Research Center for Medicinal Plant Resources, National Institutes of Biomedical Innovation, Health and Nutrition, Ibaraki 305-0834, Japan; yoshimat@nibiohn.go.jp

**Keywords:** *Eleutheococcus senticosus* leaf, memory, blood-brain barrier, dendrite, mice

## Abstract

The pharmacological properties of *Eleutherococcus senticosus* leaf have not been clarified although it is taken as a food item. In this study, the effects of water extract of *Eleutherococcus senticosus* leaves on memory function were investigated in normal mice. Oral administration of the extract for 17 days significantly enhanced object recognition memory. Compounds absorbed in blood and the brain after oral administration of the leaf extract were detected by LC-MS/MS analyses. Primarily detected compounds in plasma and the cerebral cortex were ciwujianoside C3, eleutheroside M, ciwujianoside B, and ciwujianoside A1. Pure compounds except for ciwujianoside A1 were administered orally for 17 days to normal mice. Ciwujianoside C3, eleutheroside M, and ciwujianoside B significantly enhanced object recognition memory. These results demonstrated that oral administration of the leaf extract of *E. senticosus* enhances memory function, and that active ingredients in the extract, such as ciwujianoside C3, eleutheroside M, and ciwujianoside B, were able to penetrate and work in the brain. Those three compounds as well as the leaf extract had dendrite extension activity against primary cultured cortical neurons. The effect might relate to memory enhancement.

## 1. Introduction

*Eleutherococcus senticosus* (Rupr. & Maxim.) Maxim. (synonymous with *Acanthopanax senticosus*), also known as “Siberian Ginseng” (English), “Ciwujia” (Chinese), or “Ezoukogi” (Japanese), is a species of woody shrub in the family Araliaceae [1]. The rhizomes and roots of *E. senticosus* are recorded in the Chinese and Japanese pharmacopoeias as a treatment for neurasthenia, hypertension, chronic coughing, and ischemic heart disease. In contrast, *E. senticosus* leaf is classified as food, and has been taken as tea, soups, wine, and so on. Pharmacologically, antibacterial and glucosidase inhibitory effects of *E. senticosus* leaves have been reported in in vitro experiments [2,3,4,5]. However, in vivo data of the leaf extract are hardly reported, except for reducing activity of triglycerides in high-fat diet–fed mice [6].

A phytochemical investigation revealed that the major constituents in leaves of this plant are triterpene saponins, caffeoylquinic acids, flavonoids, and polysaccharides, which are different from the rhizome constituents [7]. Therefore, we have focused on the leaf-specific effects of *E. senticosus* on neuron-related bioactivity. Our previous study showed that the water extract of *E. senticosus* leaves ameliorated amyloid β(25–35)-induced axonal atrophy using primary cultured neurons [8]. Although those results indicated neuronal activation by the leaf extract and candidates of active constituents in the extract, in vivo evidence showing the usefulness of the extract for brain function has not been obtained. This study aimed to investigate the effect of the water extract of *E. senticosus* leaves on cognitive function and determine which active constituents passed the blood–brain barrier (BBB).

## 2. Material and Methods

### 2.1. Animal Studies

All animal experiments were carried out in accordance with the Guidelines for the Care and Use of Laboratory Animals of the Sugitani Campus of the University of Toyama. All protocols were approved by the Committee for Animal Care and Use of the Sugitani Campus of the University of Toyama. The approval number for the animal experiments are A2014INM-1 and A2017INM-1. Male ddY mice (8 weeks old (Figures 1 and 6), Japan SLC, Shizuoka, Japan) and female ddY mice (6 weeks old (Figures 2–5), Japan SLC) were housed with free access to food and water and kept in a controlled environment (22 ± 2 °C, 50 ± 5% humidity, 12-h light cycle starting at 7:00 am). Mice were randomly divided to groups. The extract, a compound or vehicle solution was orally administered once a day for 17 days. 

### 2.2. Water Extract of Eleutherococcus senticosus Leaves

Fresh leaves of *E. senticosus* were collected from Fujiyoshida, Yamanashi, Japan, in June 2014. A voucher specimen (S. Isoda 201401) is kept at the Museum of Materia Medica, Institute of Natural Medicine, University of Toyama (TMPW). The dried powder of *E. senticosus* leaves (20 g) was extracted in hot water (85 °C, 200 mL) for 30 min, with this stage repeated twice. The liquid portion was then combined, filtered, and lyophilized to yield a leaf water extract. Yield of the extract was 8.62 g (43.1% of starting leaves).

### 2.3. Behavioural Test

The leaf extract was dissolved in distilled water. The leaf extract (500 mg/kg/day) or vehicle solution (distilled water) was orally administered by gavage once per day for 17 days. On day 14 of extract administration, mice were individually habituated to an open-field box composed of polyvinyl chloride (30 cm × 40 cm; height, 36 cm) for 10 min. Their paths were tracked using a digital camera system. The distance moved for 10 min was considered the locomotor activity and was analysed with Etho Vision 3.0 (Noldus, Wageningen, The Netherlands). At days 14, the object recognition test (ORT) was performed as described previously [9]. The numbers of mice in each group are 5. Ciwujianoside C3, eleutheroside M or ciwujianoside B was suspended in distilled water, and was orally administered by gavage once per day for 17 days (0.5 mg/kg/day). On days 14 of compound administration, the open-field locomotion test and ORT were performed. The numbers of mice in each group are 5. All tests were carried out in a dimly illuminated room (90 lux).

### 2.4. Absorption and Brain Penetration of Leaf Water Extract

To explore the compounds that penetrated the brain, leaf extract or vehicle solution (distilled water) was orally administered to ddY mouse (6 weeks old, female, *n* = 1) at a concentration of 5 g/kg for a single dose. Three, six and twelve hours after extract administration, mouse was euthanized and blood was collected. The plasma was obtained after centrifugation of blood at 11,000 g for 10 min at 4 °C. The brain cortex was dissected following perfusion with saline. Plasma (200 μL) was extracted with methanol, dried, and resolubilized in 100 μL methanol. The brain cortex was homogenized and extracted with methanol, dried, and resolubilized in 100 μL methanol before loading.

An Accela HPLC system (Thermo Fisher Scientific, Bremen, Germany) equipped with a quaternary pump, a built-in solvent degasser, column compartment and thermostated auto-sampler (PAL system) was adopted. The separation was achieved on an ODS-AQ column (2.0 mm × 150 mm, film thickness 3 μm) under linear gradient elution with solvents A (0.1% formic acid in water) and B (0.1% formic acid in acetonitrile). The gradient profile started with isocratic elution of 5% B for 5 min, then linearly changed to 18% B in 10 min, to 19% B in 10 min, to 37% B in 15 min, to 39% B in 7 min, to 80% B in 11 min, to 90% B in 2 min, followed by 2 min isocratic elution and a return to 5% B. The column temperature was set at 40 °C. The injection volume was 5 μL for every sample. The Accela HPLC system was hyphenated with the LTQ/Orbitrap XL mass spectrometer [10].

The MS data acquisition was achieved using an LTQ/Orbitrap XL mass spectrometer (Thermo Fisher Scientific, Waltham, MA) with an ESI probe in negative ion mode. The auto-tuned condition was determined by direct infusion of the standard chemical, ciwujianoside B, where the capillary voltage and temperature were set to −50 V and 330 °C, spray voltage to 3.0 kV, and tube lens voltage to −130 V. Sheath gas (N_2_) with a flow rate of 50 arbitrary units was used, and the auxiliary gas flow was set to 10 arbitrary units. The acquisition mass range was selected at m/z 100–2000 with a resolution of 30000. Higher collision induced dissociation (HCD)-based ESI-Orbitrap-MS/MS, collision induced dissociation (CID)-based ESI-LTQ-MSn (*n* = 1–4), and ESI-Orbitrap-MS full-scan were carried out in the MS/MS similarity networking. Both HCD and CID were conducted in data-dependent acquisition mode. The HCD energy was optimized and set to 40% normalized collision energy (NCE). The CID energy was set to 35% NCE in every stage of dissociation. All of the spectra were acquired and processed using LTQ Xcalibur and TraceFinder software (Thermo Fisher Scientific, Waltham, MA, USA) [10].

### 2.5. Isolation of Ciwujianoside C3, Eleutheroside M, Ciwujianoside B and Ciwujianoside A1 from E. Senticosus Leaves

Ciwujianoside C3, eleutheroside M, ciwujianoside B and ciwujianoside A1 were isolated from *E. senticosus* leaves as shown in our previous report [10]. Purified compounds were used as LC-MS/MS standards and for the ORT. 

### 2.6. Primary Culture

Embryos were removed from ddY mice (Japan SLC) at 14 days of gestation. The cortices were dissected, and the dura mater was removed. The tissues were minced, dissociated and grown in cultures with neurobasal medium (Invitrogen, Grand Island, NY, USA) that included 12% B-27 supplement (Invitrogen), 0.6% D-glucose and 2 mM L-glutamine on 8-well chamber slides (Falcon, Franklin Lakes, NJ, USA) coated with 5 μg/ml poly-D-lysine at 37 °C in a humidified incubator with 10% CO_2_. The seeding cell density was 1.47 × 10^4^ cells/cm^2^.

### 2.7. Measurement of Dendritic Density

For measurement of the density of dendrites, the cells were cultured for 3 days and were then treated with fresh medium containing the extract, a compound or vehicle solution (water) for 4 days. The neurons were fixed with 4% paraformaldehyde for 90 min and were immunostained with a polyclonal antibody against microtubule-associated protein 2 (MAP2, 1:2000, Abcam, Cambridge, UK) was used as a dendritic marker. Alexa Fluor 568-conjugated goat anti-rabbit IgG (1:300) was used as the secondary antibodies (Molecular Probes, Eugene, OR, USA). Nuclear counterstaining was performed using DAPI (1 μg/mL, Sigma-Aldrich). The fluorescence images were captured with a 10X objective lens using a fluorescence microscope system (Cell Observer, Carl Zeiss, Tokyo, Japan). Fifty eight to seventy two images (Figure 7A) and 51 to 105 images (Figure 7B) were captured per treatment. The lengths of the MAP2-positive dendrites were measured using a MetaMorph analyzer (Molecular Devices, Sunnyvale, CA, USA), which automatically traces and measures neurite length without measuring the cell bodies. The sum of the dendrite lengths was divided by the number of MAP2-positive neurons.

### 2.8. Statistical Analysis

Graphs were generated with GraphPad Prism 5 (GraphPad Software, La Jolla, CA, USA). One-way analysis of variance (ANOVA) with *post hoc* Dunnett’s test and repeated measures two-way ANOVA with *post hoc* Bonferroni test and were performed for statistical analysis of the data, and the statistical significance criterion *P* value was 0.05. The data are presented as the mean ± SEM.

## 3. Results

### 3.1. Leaf Extract of E. senticosus Enhances Memory Function in Mice

To investigate the effect of leaf extract of *E. senticosus* on cognitive function in young mice, the extract or vehicle solution was administered p.o. to mice for 17 days. Used dose of the leaf extract was 500 mg/kg/day. The dose was referred by other report (300 mg/kg/day) which identified brain transferred compounds in rats after oral administration of leaf extract [11]. At drug administration day 14, a locomotion test and a training session in object recognition test was performed. After a 72-h interval, a test session in the test was performed. We previously confirmed that young mice could not keep object recognition memory with an interval time longer than 24 h. As shown in Figure 1B, leaf extract treatment significantly enhanced object recognition memory in mice. Trial and drug interaction was significantly different (F(1, 8) = 28.54, *P* = 0.0007) in repeated measures two-way ANOVA, *post hoc* Bonferroni test, suggesting the leaf extract enhance object recognition memory in mice. 

On the training session day, the leaf extract was administered 1 h after a training session (day 14) and the last administration was day 16. This administration protocol suggested that the leaf extract may not temporarily stimulate memory acquisition and/or memory retention. Locomotion test showed no alteration of activity in a leaf extract-treated (Figure 1C). These data suggest that the leaf extract treatment reinforced object recognition memory. Transitions of body weights in all groups were not significantly different (Figure 1D).

### 3.2. Leaf Extract-Derived Compounds Penetrate the BBB after the Oral Administration of the Extracts

To clarify active principles of the leaf extract, we tried to detect the leaf extract-derived compounds that are delivered into the brain. Using the high-accuracy quasi-molecular ion ([M+HCOO]^−^) and a mass error of ±1 mmu, we comprehensively characterized 4 compounds by comparing their MS-MS data and fragmentation patterns with those of reference standards or reported compounds. Then, we profiled the chemicals that were delivered into the blood and brain in ddY mice with HPLC-FT-MS. Ciwujianoside C3 (MW 1059.25) was detected in plasma and the cerebral cortex at 3 h after administration (Figure 2A,B). Eleutheroside M (MW 1205.39) (Figure 3A,B), ciwujianoside B (MW 1189.35) (Figure 4A,B) and ciwujianoside A1 (MW 1221.39) (Figure 5A,B) were detected in plasma at 3, 6 and 12 h after, and in the cerebral cortex at 6 h after administration. Those 4 compounds were not detected in plasma and cerebral cortex of vehicle solution-treated mice. Data were summarized in Table 1. Usually higher doses than effective doses are selected to detect the metabolites of herbal extracts [12], formulations [13], and even single compounds [14,15] in biosamples. Therefore, we orally administered a high dose of the leaf extract (5 g/kg) to ddY mice.

### 3.3. Brain Penetrated Compounds Enhances Memory Function in Mice

To investigate the effect of ciwujianoside C3, eleutheroside M and ciwujianoside B on cognitive function in young mice, the extract or vehicle solution was administered p.o. to mice for 17 days. Among the four compounds identified as having penetrated into the brains of the mice, ciwujianoside A1 was not isolated in sufficient amounts to allow for in vivo experimentation. Used dose of compounds were set as follows. Seo et al. reported contents of ciwujianoside C3 and ciwujianoside B in leaves were 0.0105% and 0.0421%, respectively [16]. Since yield of the leaf extract was 43.2% used in this study, suspected contents of ciwujianoside C3 and ciwujianoside B were expected as 0.024% and 0.098%, respectively. There is no information about the content of eleutheroside M in leaves. Therefore, we supposed contents of ciwujianoside C3, eleutheroside M and ciwujianoside B in the leaf extract might be approximately 0.1%. Since an effective dose of the leaf extract was 500 mg/kg/day (Figure 1A), 0.1% of the dose, 0.5 mg/kg/day was set as an administration dose of compounds. 

At drug administration day 14, a locomotion test and a training session in object recognition test was performed, and after a 72 h interval, a test session in the test was performed. As shown in Figure 6B, any treatment of ciwujianoside C3, eleutheroside M and ciwujianoside B significantly enhanced object recognition memory in mice. Trial and drug interaction was significantly different between vehicle treatment and ciwujianoside C3 (F(1, 8) = 19.18, *P* = 0.0024), vehicle treatment and eleutheroside M (F(1, 8) = 5.681, *P* = 0.0443), vehicle treatment and ciwujianoside B (F(1, 8) = 30.59, *P* = 0.0006) in repeated measures two-way ANOVA, *post hoc* Bonferroni test. Locomotion test showed no alteration of activity by compounds’ administration (Figure 6C). These data suggest that those 3 compounds treatment reinforced object recognition memory. Transitions of body weights in all groups were not significantly different (Figure 6D).

### 3.4. Leaf Extract and Brain Penetrated Compounds Enhance Dedrite Growth in Cortical Neurons

We previously found that a extract enhancing dendritic growth in cultured cortical neurons increased in object recognition memory in mice [17]. Therefore, dendritic growth activities of the leaf extract, ciwujianoside C3, eleutheroside M and ciwujianoside B were investigated. The leaf extract at a dose of 1 μg/ml (Figure 7A), and three compounds at doses of 1 and 10 μM (Figure 7B) significantly increased dendrite lengths in cultured cortical neurons.

## 4. Discussion

We found for the first time that leaf extract of *E. senticosus* upregulates cognitive function in young mice. Four compounds passed into the brain were also identified after oral administration of the extract. Since ciwujianoside C3, eleutheroside M, ciwujianoside B and ciwujianoside A1 are contained in the extract, those compounds are possibly absorbed in blood in native forms and transferred to the brain without metabolization. Our data showed that at least ciwujianoside C3, eleutheroside M and ciwujianoside B enhanced memory function when applied per oral route (Figure 6A). These data indicate that ciwujianoside C3, eleutheroside M and ciwujianoside B are active principles in the leaf extract for neuronal activation.

Wang et al. identified the absorbed components in rat plasma and brain after oral administration of leaf extract of *E. senticosus* [11]. In the report, triterpenoid saponins involving ciwujianoside C3 and ciwujianoside B were detected in plasma, but not in the brain. Chlorogenic acid, rutin, hyperoside and so on, which are not specific compounds in *E. senticosus*, were detected in the brain in the report. If speculating different pharmacokinetic results, BBB transporting system might be different from mice and rats. Perhaps, contents of triterpenoid saponins was different from our extract and the leaf extract used by Wang et al. Because their extract was extracted by 60% ethanol but not water. Glycosides like ciwujianoside C3, eleutheroside M and ciwujianoside B might not extracted well in 60% ethanol.

Our previous study performed in primary cultured cortical neurons showed that the aqueous extract of *E. senticosus* leaves exerted axonal growth activity in amyloid β-induced degeneration model [8]. Subsequent bioassay-guided fractionation afforded seven new oleanane-type triterpene saponins, ezoukoginosides A−G as potent components for axonal growth effect [8]. However, those compounds have not detected in the brain after oral administration of the leaf extract in this study. 

In contrast, the present data indicates that specific compounds contained in leaves of *E. senticosus* play a critical role for the extract-revealed memory enhancement. As we reported in other studies, active compounds indicated by bioassay-guided fractionation in cellular experiments [18] do not necessary correspond to real active principles transferred in the brain [19]. Especially in compounds working in the brain, BBB penetration is very critical. Therefore, it is meaningful that at least 3 compounds were identified as brain transferred memory enhancers. In addition, our previous chemical study revealed that contents of ciwujianoside C3, eleutheroside M and ciwujianoside B in *E. senticosus* leaves were different in cultivated areas [10]. Leaves with high contents of those compounds might be good quality ones as nutritional sources for memory enhancement.

Very interestingly, ciwujianoside C3, eleutheroside M and ciwujianoside B passed BBB in spite of large of molecular weight (more than 1000) and hydrophilicity. For example, a hydrophilic compound, morphine-6-glucuronide are considered to be actively transported in the brain by unknown transporter [20]. There might be something transport system for ciwujianoside C3, eleutheroside M and ciwujianoside B. Pharmacological effects of ciwujianoside C3, eleutheroside M and ciwujianoside B were hardly reported except for following two reports using damaged cell conditions; ciwujianoside C3 inhibits proinflammatory cytokine levels in lipopolysaccharide (LPS)-stimulated RAW 264.7 cells [21], ciwujianoside B reduces DNA damage in bone marrow cells exposed to radiation [22]. No studies reported effects of these three compounds on neuronal function in healthy condition. The present study indicated the leaf extract and three compounds extended dendrites in cultured cortical neurons (Figure 7A,B). As we reported in other report [17], substances having dendritic growth activity in cultured neurons often corelates to memory enhancement activity in vivo. For unraveling signaling pathways of ciwujianoside C3, eleutheroside M and ciwujianoside B involving in dendritic growth and memory enhancement, we are now identifying direct binding molecules of ciwujianoside C3, eleutheroside M and ciwujianoside B using DARTS method as we succeeded [19]. As a next step, detail analyses of signaling mechanisms as well as pharmacokinetics of those compounds should be forwarded.

We are interested in getting evidences of cognitive enhancement of the leaf extract of *E. senticosus* as an attractive nutritional source for cognitive function. In near future, we will complete the first clinical trial performed in healthy subjects.

## 5. Conclusions

The present study demonstrated memory enhancement by oral administration of the leaf extract of *E. senticosus*, and active principles in the extract penetrating and working in the brain. Elucidating how those compounds act on neuronal excitation must give new insights on regulatory mechanism of the cognitive function.

## Figures and Tables

**Figure 1 nutrients-11-01142-f001:**
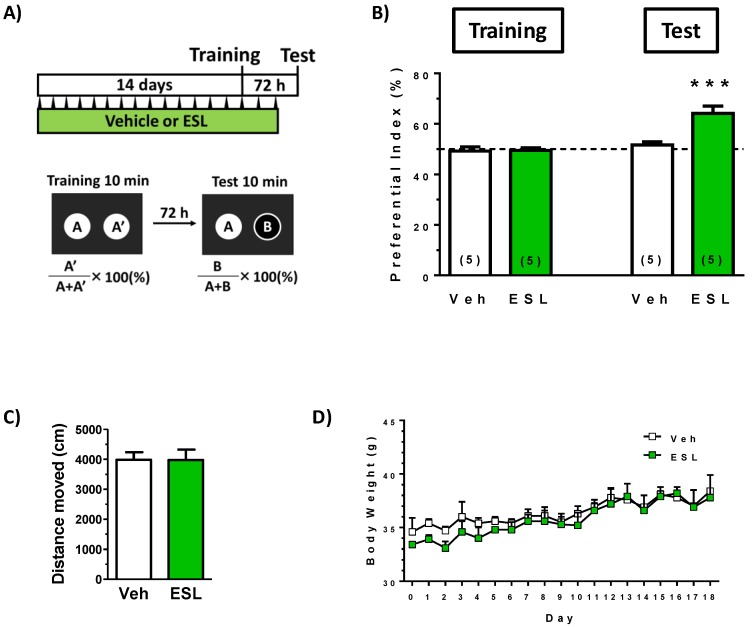
Effects of the leaf extract of *E. senticosus* on object recognition memory in mice. (**A**) The leaf extract of *E. senticosus* (500 mg/kg, p.o.) (ESL) or vehicle solution (Veh) was administered for 17 days to mice (males, 8 weeks old). On administration day 14, an object recognition test was started. (**B**) The preference indices of the training and test sessions are shown. Above 50% of the preferential index in a test session shows memory enhancement. (*** *P* < 0.001, repeated measures two-way ANOVA, *post hoc* Bonferroni test; *n* = 5 mice). (**C**) Locomotion was evaluated as moved distances for 10 min in an open field box on day14. Statistical analysis was done by one-way ANOVA. *P* > 0.05. (**D**) Body weights in two groups were measured and analyzed by repeated measures two-way ANOVA. *P* > 0.05.

**Figure 2 nutrients-11-01142-f002:**
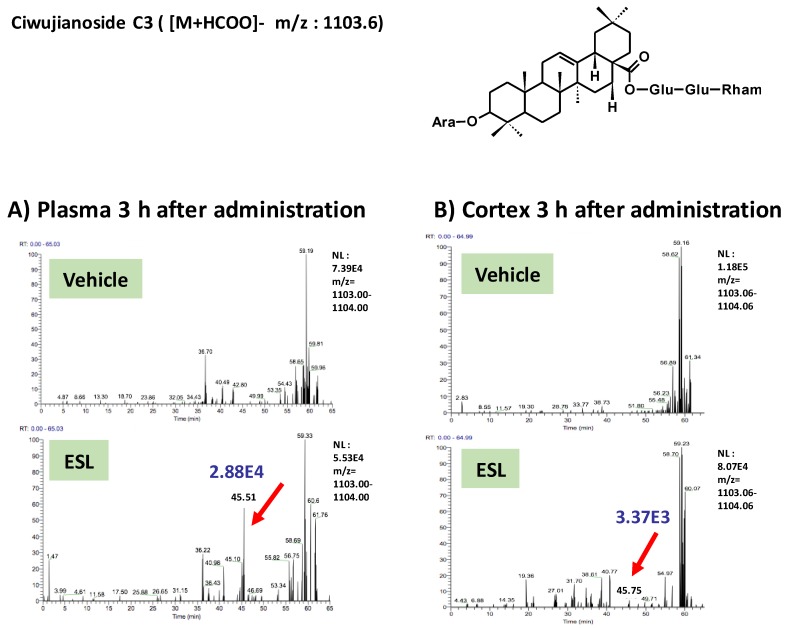
Ciwujianoside C3 was detected in plasma and the cerebral cortex after oral administration of the leaf extract of *E. senticosus.* A structure of ciwujianoside C3 and LC-MS charts of plasma (**A**) and cerebral cortex (**B**) from vehicle-treated and leaf extract-treated mice 3 h administration of the extract. Red arrows indicate a peak of ciwujianoside C3. A blue value means intensities of the peak.

**Figure 3 nutrients-11-01142-f003:**
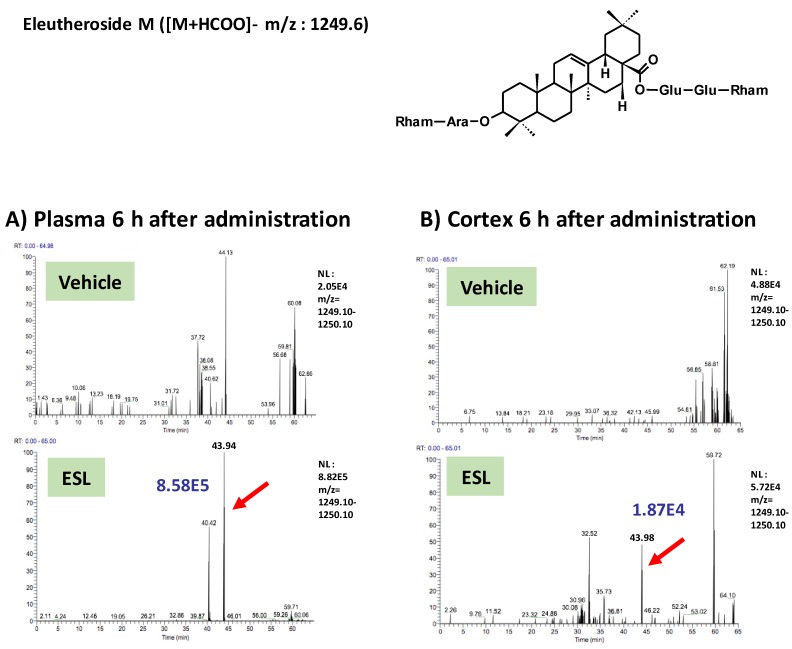
Eleutheroside M was detected in plasma and the cerebral cortex after oral administration of the leaf extract of *E. senticosus.* A structure of eleutheroside M and LC-MS charts of plasma (**A**) and cerebral cortex (**B**) from vehicle-treated and leaf extract-treated mice 6 h administration of the extract. Red arrows indicate a peak of eleutheroside M. A blue value means intensities of the peak.

**Figure 4 nutrients-11-01142-f004:**
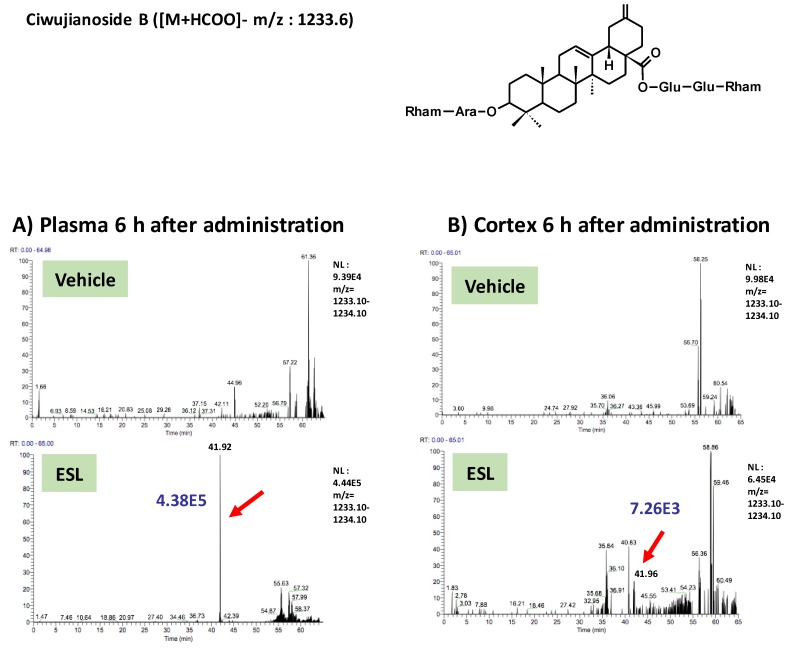
Ciwujianoside B was detected in plasma and the cerebral cortex after oral administration of the leaf extract of *E. senticosus.* A structure of ciwujianoside B and LC-MS charts of plasma (**A**) and cerebral cortex (**B**) from vehicle-treated and leaf extract-treated mice 6 h administration of the extract. Red arrows indicate a peak of ciwujianoside B. A blue value means intensities of the peak.

**Figure 5 nutrients-11-01142-f005:**
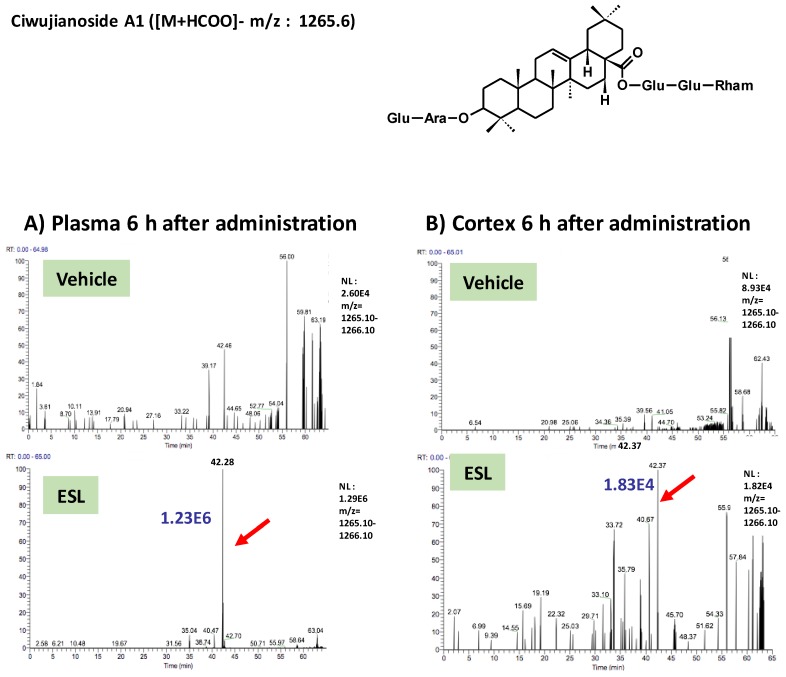
Ciwujianoside A1 was detected in plasma and the cerebral cortex after oral administration of the leaf extract of *E. senticosus.* A structure of ciwujianoside A1 and LC-MS charts of plasma (**A**) and cerebral cortex (**B**) from vehicle-treated and leaf extract-treated mice 6 h administration of the extract. Red arrows indicate a peak ciwujianoside A1. A blue value means intensities of the peak.

**Figure 6 nutrients-11-01142-f006:**
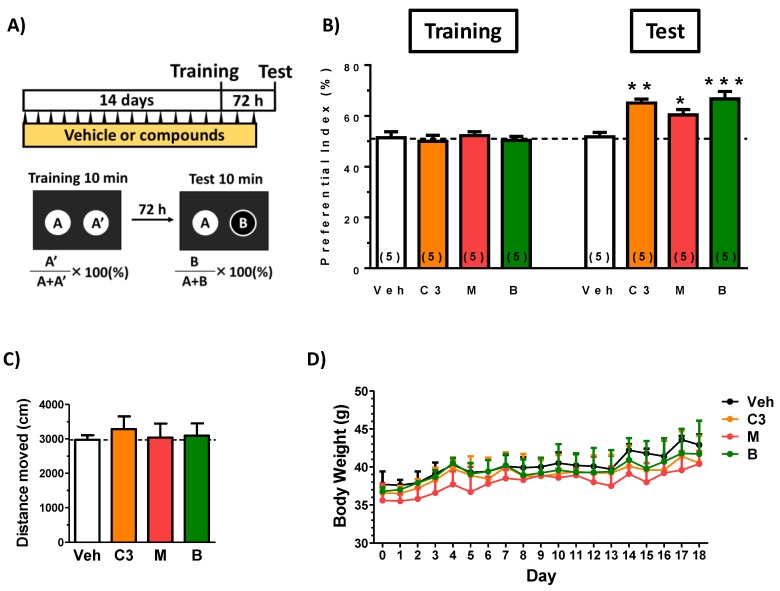
Effects of ciwujianoside C3, eleutheroside M and ciwujianoside B on object recognition memory in mice. (**A**) Ciwujianoside C3 (C3), eleutheroside M (M), ciwujianoside B (B) (each 0.5 mg/kg, p.o.) or vehicle solution (Veh) was administered for 17 days to mice (males, 8 weeks old). On administration day 14, an object recognition test was started. (**B**) The preference indices of the training and test sessions are shown. (* *P* < 0.05, ** *P* < 0.01, *** *P* < 0.001, repeated measures two-way ANOVA, *post hoc* Bonferroni test; *n* = 5 mice). Above 50% of the preferential index in a test session shows memory enhancement. (**C**) Locomotion was evaluated as moved distances for 10 min in an open field box on day14. Statistical analysis was done by one-way ANOVA. *P* > 0.05. (**D**) Body weights in four groups were measured and analyzed by repeated measures two-way ANOVA. *P* > 0.05.

**Figure 7 nutrients-11-01142-f007:**
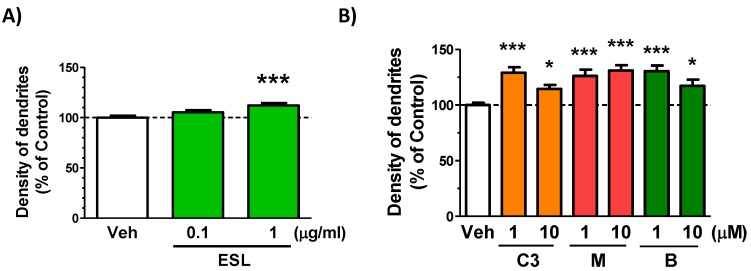
Effect of leaf extract and brain penetrated compounds on dedrite growth in cortical neurons. (**A**) Cortical neurons were cultured for three days and then treated with or without the leaf extract (ESL, 0.1 and 1 μg/mL). Four days after treatment, the cells were fixed and immunostained for MAP2. The density of MAP2-positive dendrites was quantified for each treatment (*** *P* < 0.0001, one-way ANOVA with post hoc Bonferroni test, *n* = 58–72). (**B**) Cortical neurons were cultured for three days and then treated with or without ciwujianoside C3 (C3), eleutheroside M (M), ciwujianoside B (B) (each 1 and 10 μM). Four days after treatment, the cells were fixed and immunostained for MAP2. The density of MAP2-positive dendrites was quantified for each treatment (* *P* < 0.05, *** *P* < 0.0001, one-way ANOVA with post hoc Bonferroni test, *n* = 51–105).

**Table 1 nutrients-11-01142-t001:** Summary of the detected compounds in the plasma and the cerebral cortex of mice after oral administration of the leaf extract of *E. senticosus*.

Detected Components	Plasma	Cerebral Cortex
3 h	6 h	12 h	3 h	6 h	12 h
ciwujianoside C3	√			√		
eleutheroside M	√	√	√		√	
ciwujianoside B	√	√	√		√	
ciwujianoside A1	√	√	√		√

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
