# Peer review of "Memory Enhancement by Oral Administration of Extract of Eleutherococcus senticosus Leaves and Active Compounds Transferred in the Brain"

_nutrients, 2019, doi:10.3390/nu11051142_

Round 1
Reviewer 1 Report
1. The current discussiondoes not mention the relevant literature in the main functional components (ciwujianoside C3, eleutheroside M, ciwujianoside B and ciwujianoside A1). It is difficult to convince the reader (at least to provide the relevant mechanism for the improvement of brain function). Thus it should be discussed in detail that how ciwujianoside C3, eleutheroside M, ciwujianoside B and ciwujianoside A1 may increase brain function
2. In order to meet the standards published in this issue, the author should be required to provide one or two mechanism diagrams (the performance of key proteins) to make this story complete.
Author Response
Thank you for prompt reviewing and suggesting important points to improve our paper. We revised according your comments as follows. Changed parts in the text are blue marked.
1. The current discussiondoes not mention the relevant literature in the main functional components (ciwujianoside C3, eleutheroside M, ciwujianoside B and ciwujianoside A1). It is difficult to convince the reader (at least to provide the relevant mechanism for the improvement of brain function). Thus it should be discussed in detail that how ciwujianoside C3, eleutheroside M, ciwujianoside B and ciwujianoside A1 may increase brain function
Response
As mentioned in previous draft, brain functions of ciwujianoside C3, eleutheroside M and ciwujianoside B have never been reported yet. To suppose it, we added dendrite extension data of those compounds and the leaf extract as new Figure 7. Data showed the extract and compounds significantly enhanced dendritic density in cultured cortical neurons (line 242 – 258).
2. In order to meet the standards published in this issue, the author should be required to provide one or two mechanism diagrams (the performance of key proteins) to make this story complete.
Response
As you pointed, it is important to identify involving molecules in the signaling pathway of active principles. We consider that the best approach is to identify direct binding molecules of those compounds in neurons. That challenge is now on going, and results will be reported in future. At the present, our paper focuses on the leaf extract of E. senticosus as an attractive nutritional source for cognitive function. It is possibly meaningful for readers of Nutrients.
Response
Considering totally your advice, we added new discussion as follows in line 295 – 307.
Pharmacological effects of ciwujianoside C3, eleutheroside M and ciwujianoside B were hardly reported except for following two reports using damaged cell conditions; ciwujianoside C3 inhibits proinflammatory cytokine levels in LPS-stimulated RAW 264.7 cells [19], ciwujianoside B reduces DNA damage in bone marrow cells exposed to radiation [20]. No studies reported effects of these three compounds on neuronal function in healthy condition. The present study indicated the leaf extract and three compounds extended dendrites in cultured cortical neurons (Figure 7A and 7B). As we reported in other report [15], substances having dendritic growth activity in cultured neurons often corelates to memory enhancement activity in vivo. For unraveling signaling pathways of ciwujianoside C3, eleutheroside M and ciwujianoside B involving in dendritic growth and memory enhancement, we are now identifying direct binding molecules of ciwujianoside C3, eleutheroside M and ciwujianoside B using DARTS method as we succeeded [17]. As a next step, detail analyses of signaling mechanisms as well as pharmacokinetics of those compounds should be forwarded.

Reviewer 2 Report
Procedures for isolation and purification of ciwujianoside B, ciwujianoside C3 and eleutheroside M must be presented.
Author Response
To referee 1
Referee’s comment
Procedures for isolation and purification of ciwujianoside B, ciwujianoside C3 and eleutheroside M must be presented.
Response
Thank you for important suggestion. We add a method of isolation of compounds in “2.5. Isolation of Ciwujianoside C3, Eleutheroside M, Ciwujianoside B and Ciwujianoside A1 from E. senticosus leaves” in line 106 – 110. Changed parts in the text are yellow marked.

Reviewer 3 Report
Dear authors,
Please carefully read the attached file and revised your manuscript in accordance to the comments.
1- Page 1, line 1: The title is changed to: Memory enhancement effect of Eleutherococcus senticosus aqueous extract and their active principles in mice.
2- Page 1,line 13: E. senticosus
3- Page 1, line 19: Please use the same words for example active principles, Active constituents, Active compounds, or pure compounds, the different words could confuse the readers.
4- Page 1, line 25: Keywords.
5- Page 2,Material and method section:
- Please clearly mention the inclusion and exclusion criteria.
- Please clearly mention the normal mice characteristics.
- Please create the experimental design subtitle and clearly mentioned your experimental groups.
- Please clearly mention the total numbers of animals and the numbers of animals in each group.
- Page 2, Line 47.please write animals whereas animal studies.
- Page 2, Line 61.43.1% w/w dray mater? Please mention.
- Page 2, Line 74: What is your rationale for using these doses? Any criteria? Please mention.
- Page 2, Line 89: please mention the valid reference numbers.
6- Page 3, line 104, it is better you firstly mentioned the one way ANOVA.
7- Page 5, line 151, the extract.
8- Page 7, line 172, in the horizontal description of the table, and please write the active components.
9- Page 8,line 209,per oral route.
10- Page 8, in your discussion; compare your results with more and more similar studies.
11- The non-significant statics should me mentioned by: p>0.001.
12- The hole of the abbreviations’ should be described completely in the first use.
13- All of the figures are unclear. Please use the more colorful and clearer figures. Please clearly mention. Please clearly indicate the vertical and horizontal coordinates of the figures.
14- The figures and tables should be stand alone. Please give them a thorough description of them (including abbreviations).
15- Please mention the applied conclusion for future studies.
With best wishes
Journals referee
Author Response
To referee 2
Thank you for prompt reviewing and suggesting many important points to improve our paper. We revised one by one according your comments as follows. Changed parts in the text are yellow marked.
1- Page 1, line 1: The title is changed to: Memory enhancement effect of Eleutherococcus senticosus aqueous extract and their active principles in mice.
Response
The title has been changed to “Memory enhancement effect of Eleutherococcus senticosus leaf aqueous extract and active principles in mice”.
2- Page 1,line 13: E. senticosus
Response
Changed to E. senticosus after 2nd appearance.
3- Page 1, line 19: Please use the same words for example active principles, Active constituents, Active compounds, or pure compounds, the different words could confuse the readers.
Response
The word of “active principles” were used thoroughly in the text.
4- Page 1, line 25: Keywords.
Response
Typo was corrected.
5- Page 2,Material and method section:
- Please clearly mention the inclusion and exclusion criteria.
Response
In the animal study using same age and same sex mice, inclusion and exclusion criteria are not generally mentioned. However, we add sentences in the method as “Mice were randomly divided to groups. The extract, a compound or vehicle solution was orally administered once a day for 17 days.” in line 54 – 56.
- Please clearly mention the normal mice characteristics.
Response
As you felt unnatural, the word of “normal” was strange expression. The word of “normal” was deleted.
- Please create the experimental design subtitle and clearly mentioned your experimental groups.
Response
Already we mentioned experimental design in sections 2.3. Behavioural Test, 2.4. Absorption and Brain Penetration of Leaf Water Extract, Figure 1A and Figure 6A. In addition, we have newly mentioned “Mice were randomly divided to groups. The extract, a compound or vehicle solution was orally administered once a day for 17 days.” in line 54 – 56.
- Please clearly mention the total numbers of animals and the numbers of animals in each group.
Response
Although we already mentioned used animal numbers in each figure legends, a sentence has been added in line 71 – 72.
- Page 2, Line 47.please write animals whereas animal studies.
Response
We added information of animals in line 52 – 53.
- Page 2, Line 61.43.1% w/w dray mater? Please mention.
Response
We corrected the expression to “Yield of the extract was 8.62 g (43.1% of starting leaves).” in line 62 -63.
- Page 2, Line 74: What is your rationale for using these doses? Any criteria? Please mention.
Response
We added the reason why we chose 500 mg/kg/day in line 120 -121. The dose was referred by other report (300 mg/kg/day) which identified brain transferred compounds in rats after oral administration of leaf extract [10].
- Page 2, Line 89: please mention the valid reference numbers.
Response
We carefully corrected and added references and those numbering.
6- Page 3, line 104, it is better you firstly mentioned the one way ANOVA.
Response
We mentioned firstly one-way ANOVA in line 112 – 113.
7- Page 5, line 151, the extract.
Response
We corrected typos throughout text.
8- Page 7, line 172, in the horizontal description of the table, and please write the active components.
Response
We revised Table 1 (line 183 – 184).
9- Page 8,line 209,per oral route.
Response
We added “route” in line 224.
10- Page 8, in your discussion; compare your results with more and more similar studies.
Response
We added discussions in line 227 -249.
11- The non-significant statics should me mentioned by: p>0.001.
Response
Non-significant statics have been shown as P > 0.05 in lines 142, 144, 216 and 217.
12- The hole of the abbreviations’ should be described completely in the first use.
Response
Spelling out was done in the first use of abbreviations.
13- All of the figures are unclear. Please use the more colorful and clearer figures. Please clearly mention. Please clearly indicate the vertical and horizontal coordinates of the figures.
Response
All figures and table were replaced to clear ones. To help understanding y-axis value, explanations have been added in lines 139 – 140, 214 – 215.
14- The figures and tables should be stand alone. Please give them a thorough description of them (including abbreviations).
Response
Sub figures have been captioned correctly and independently, and corresponded with descriptions in the text.
15- Please mention the applied conclusion for future studies.
Response
Future direction has been added in Conclusions as “Elucidating how those compounds act on neuronal excitation must give new insights on regulatory mechanism of the cognitive function.” in line 265 – 267.

Round 2
Reviewer 1 Report
To review the revised version, the text presentation was clear and concise. I encourage this manuscript to be published.Reviewer 2 Report
publish in current form
Reviewer 3 Report
Dear authors,
Page 2,line 72.The numbers of mice in each group were 5.
Page 2,line 75.Please delete the sentences (The 75 numbers of mice in each group are 5)With Best Wishes,
Journals referee